# Impact of built environment on residential online car-hailing trips: Based on MGWR model

Yan Cao[1,2,3], Yongzhong Tian[1,2,3]*, Jinglian Tian[1,2,3], Kangning Liu[4], Yang Wang[5]

1 Chongqing Jinfo Mountain Karst Ecosystem National Observation and Research Station, School of Geographical Sciences, Southwest University, Chongqing, China, 2 Chongqing Engineering Research Center for Remote Sensing Big Data Application, School of Geographical Sciences, Southwest University, Chongqing, China, 3 Daotian Science and Technology Limited Company, Chongqing, China, 4 Chongqing Geomatics and Remote Sensing Center, Chongqing, China, 5 Chongqing Soil and Water Conservation Monitoring Centre, Chongqing, China

* tyzlf@swu.edu.cn

## Abstract

With the development of smart mobile devices and global positioning technology, people's daily travel has become increasingly dependent on online car-hailing. Meanwhile, it has also become possible to use multi-source data to explore the factors influencing urban residents' car-hailing trips. Using online data on car-hailing trajectories, points of interest (POIs) data and other auxiliary data, the paper explores how the built environment impacts online car-hailing passengers. Within a 200 x 200m research grid, the unique spatiotemporal patterns of weekday car-hailing trips during a one-week period are analyzed, using statistics on pick-ups and drop-offs at different time of the day. By combining these data with built environment variables and various economic and traffic indicators, a multi-scale geographically weighted regression (MGWR) model is developed for different time scales. The MGWR model outperforms the classical geographically weighted regression (GWR) model and the ordinary least squares (OLS) regression model in terms of goodness of fit and all other aspects. More importantly, this study finds a high degree of temporal and spatial heterogeneity in the impact of built environment factors on local car-hailing trips across different regions, and the paper analyzes the business residence coefficient in detail. The study provides valuable insights to help improve the level of urban transportation services, as well as urban transportation planning and construction.

## 1. Introduction

The distributions of human trajectories in time and space is highly regular, and is notable by the independence of each person's travel time and distance while also meeting the trajectories of others at frequent locations [1]. Various spatiotemporal phenomena are caused by the spatial redistribution of individuals' dynamics at the geographic scale [2]. The choice of travel mode is an important factor affecting urban planning, transportation construction and

**Data Availability Statement:** This paper contains information from OpenStreetMap and OpenStreetMap Foundation, which is made available under the Open Database License (https://www.openstreetmap.org/copyright). Vehicle

trajectory data are acquired from the GAIA open data initiative (https://gaia. didichuxing.com) of DiDi Chuxing; POI data are acquired from the open platform of Gaode Map (https: //lbs.amap.com); Population data are acquired from the WorldPop project (https://www.worldpop.org/); Housing price data are acquired from the Lianjia (https://cd.lianjia. com/).

**Funding:** This research was funded by the State Key Programme of National Social Science Foundation of China, No. 18AJY018 and Chongqing Social Science Planning Project, No.2020PY28. These fundings provided some important experimental data and financial support for this study.

**Competing interests:** The authors have declared that no competing interests exist.

forecasting, resource flow, and epidemic forecasting. Consequently, it has a significant impact on the productive life of residents [3, 4]. Meanwhile, as urbanization accelerates in developing countries, the changing urban built environment significantly affects the travel demand and patterns of urban residents. To promote the efficient flow of urban resources, improve urban planning and construction, and alleviate the problem of urban diseases, there is a growing interest among scholars in the correlations between urban built environment factors and travel demand. The findings of these studies are expected to serve as a foundation for making decisions about efficient, healthy and green urban development [5–7]. At the same time, with advances in global positioning technology, information science, and communication technology, the generation of location based service(LBS) data has led scholars to pay more attention to the spatial and temporal characteristics of residential trips [8, 9].

Studying the relationship between online car-hailing and the urban built environment is an important but complex task. First, the influences associated with the demand for car-hailing are often influenced by a combination of internal and external objective factors, and in addition to the internal factors of drivers and residents themselves, the external factors are difficult to elucidate according to the different functional zoning for urban land use. Second, general linear regression models can only capture the global characteristics of regions. In addition to global characteristic regions, the local characteristics of variables should also be considered in the study of spatio-temporal data, that is, the scales of action of different variables in different local regions should also be considered in order to increase the reliability and authenticity of the models. Third, the demand for car-hailing and the explanatory variables also have significant time variation, for example, residents have different trip demand characteristics at different time of the day and different days of a week, and the explanatory variables also exist in both business and non-business states at different time of the day.

The contribution of this study is mainly reflected in the following three areas. Specifically, the unique pattern of weekday online car-hailing trips is described spatially and temporally by constructing a small-scale research grid as a research unit based on GPS data from DiDi. Next, the number of POIs in the study area is captured by Python and combined five categories of indicators, namely road density, urban transport facilities, population density, average housing price and diversity of regional facilities, as a system of explanatory variables for the built environment. And through the method of spatial analysis, the explanatory variables within each grid are counted to achieve the spatialisation of the explanatory variables. Finally, this paper combining the temporal characteristics of residents' trips and the explanatory variables, the MGWR model is developed for the characteristic time periods of residents' trips. The model verifies the significant spatial and temporal correlation between residents' trips and the built environment and the obvious spatial heterogeneity between different explanatory variables.

Our work helps to improve the supply-demand balance of urban online car-hailing operations and provides a reference for online car-hailing companies to develop more reasonable passenger-finding strategies. This study also aims to provide a basis for decision-making to alleviate urban congestion challenges, improve the efficiency of urban public transport operations and service levels, and improve the planning and construction of transport and cities. The remainder of this paper is as follows. Section 2 summarizes how the built environment influences car-hailing trips as well as the spatial heterogeneity of those factors. Section 3 outlines the study area, research data and methods used in this article, including online car-hailing trips data, built environment variables and MGWR model. Section 4 and section 5 introduce the model result and discussion. Section 6 summarizes the conclusions. Section 7 provides the limitations and future work.

## 2. Literature review

The built environment is a driving force for travel demand. Indeed, previous studies have shown the built environment has a significant impact on residents' demand for car-hailing trips, and there are significant differences between different modes of transportation such as buses, taxis, and bicycles [10]. In this paper we examine the spatio-temporal correlation between the built environment and passengers using online car-hailing services, and summarize earlier related studies in terms of how the built environment influences car-hailing trips as well as the spatial heterogeneity of those factors.

The built environment is usually defined as a term that encompasses a wide range of different spatial features in the environment that generate the demand for taxi rides [11]. Previous studies have found a strong influence of the built environment on car-hailing trips. Cervero and Ewing [10, 12], proposed five basic categories of key variables for the design of explanatory variables in the analysis of correlation between built environment and car-hailing trips. These can be summarized in the "5D" principles of density, diversity, design, destination accessibility, and travel distance. These built environment characteristics have been represented in the literature by several variables. Strongly mixed land uses may lead to increasing demand for taxis, with different types of land uses, such as finance, residential, office and land use diversity, having an impact on car-hailing trips [10, 13]. The POIs data obtained from the navigation software can provide more accurate location information for land use type and travel activities [14, 15]. Population density and accessibility to public transportation are often used as predictors of car-hailing trips [16–18]. The density of the road network in the study area has been used for travel behavior analysis as an objective condition limiting traffic trips [19, 20]. Housing prices, to some extent, reflect income levels and have been used as a proxy for socioeconomic status [11]. Additionally, there exist unique needs for taxis in airports, train stations, and bus stations, which serve as external transportation hubs [21, 22].

Ordinary least squares (OLS) is the most commonly used regression analysis model [23, 24]. However, all variables in the OLS model are assumed to be spatially smooth and there is no spatial heterogeneity across the study area [25]. From the "Law of Spatial Heterogeneity", it is clear that the structure and regional characteristics of the city have a strong influence on the number of taxi passengers in the area, such that the effect of the explanatory variables may be highly spatially heterogeneous [26]. Geographically weighted regression (GWR), as one of the extended models of OLS, can better solve the "spatial non-stationarity" in the regression of geographic data, and capture local variance in the data [14, 26, 27]. GWR model i also commonly used in studies related to the urban heat island effect, air pollution, urban sprawl etc. [28–30].

In comparison with OLS, the GWR model is better suited for revealing the characteristics of taxi trip influencing factors than OLS, but its explanatory power of spatial heterogeneity within a local area is limited. Few previous studies have addressed the differences in scale of spatial heterogeneity of different influencing factors, despite this task being a basic aspect of geographic research. Geographic retrieval and data analysis are primarily about scale, according to Goodchild and McMaster [31, 32]. Current research methods do not address problems of scale effectively: semi-parametric geographically weighted regression (SGWR) and geographically weighted logistic regression (GWLR) can provide partial solutions to global and local scale problems [16, 30, 33], but different variables can only be classified into global and local categories, and their impacts cannot be subdivided further. To address this weakness, Fotheringham proposed the MGWR method in 2017. Yu et al. in 2019 refined the statistical inference method for MGWR, offering the possibility of the general application of MGWR in empirical studies [23, 34, 35].

MGWR improves on the traditional GWR in the following ways. First, it allows for varying amounts of spatial smoothing of each variable, addressing the flaws of the classic GWR. The second point is that each variable's bandwidth represents the spatial scale at which that spatial process occurs. Third, the model's multi-bandwidth method better represents the reality of spatial processes and their effectiveness.

In addition to the lack of research on the scale of spatial heterogeneity, existing studies have some limitations as follows. First, most studies are based on traditional vehicle GPS or station statistics due to privacy concerns [33, 36, 37]. These data are difficult to obtain and have low positioning accuracy. Taxi trajectory data are used in this study. Taxis are becoming an increasingly popular mode of travel and playing a vital role in the mobility of the built environment due to their wide coverage, convenience and accessibility, especially for the online car-hailing represented by DiDi, which are more accessible and convenient [38, 39]. With DiDi's large amount of trajectory data and data representation, it offers the possibility to explore the spatial and temporal characteristics of residents' trips. Second, the explanatory variables of the built environment are mostly derived from traditional surveys or official statistics, and the existing research scales are mostly limited to the kilometer grid or street scale, hindering the application of smaller research scales. Third, the explanatory variables are all assigned a single fixed bandwidth scale, such that spatial heterogeneity cannot be fully accounted for, making it difficult to further improve the model fitting accuracy.

How to explore the spatio-temporal characteristics of residential online car-hailing trips under a small research scale, as well as the specific influence of built environment variables on residential trips and spatial heterogeneity characteristics is an important issue that cannot be ignored. In this paper, Origin-Destination (OD) information is extracted based on DiDi order trajectory data, and the spatio-temporal characteristics of residents' DiDi trips in different periods are analyzed on a small-scale research grid. Detailed POIs data and other auxiliary variables, such as housing prices, road network and population from relevant open website apis, are used as factors influencing the built environment on residential trips. The influencing factors are put into the MGWR model to verify the significant relationship between residential online car-hailing trips and built environment, the obvious spatial heterogeneity among different explanatory variables is discussed in detail.

# 3. Materials and methods

## 3.1. Study area

Chengdu (102˚54′E~104˚53′E, 30˚05′N~31˚26′N) is located in southwest China, in the western part of the Sichuan Basin and at the eastern edge of the Qinghai-Tibet Plateau. Chengdu is an important central city in western China, the core city of the Chengdu-Chongqing economic circle, as well as the capital of Sichuan Province. Chengdu's unique geographical location makes it one of the most sought-after residential cities in China. At the end of 2021, the population of Chengdu was 21.19 million and its GDP was 199.168 billion yuan. The central city of Chengdu typically has dense economic and industrial activity, rich and diverse internal land use types, and a high number of car-hailing trips. Our study area is located within Chengdu's third ring road, and it's about 200 square kilometers, containing six administrative districts: Qingyang, Jinniu, Chenghua, Jinjiang, Wuhou and Longquanyi (Fig 1).

To accurately analyze the characteristics of online car-hailing trips at small regional scales, previous studies usually divides regions into many small cells, such as postal code tabulation areas (ZCTA) and census block groups (CBG) [23, 26]. However, an overly large research grid does not match the real travel conditions of residents' trips and tends to result in a lack of realism in the model results. Our experiments show that the spatial pattern of DiDi trips can be

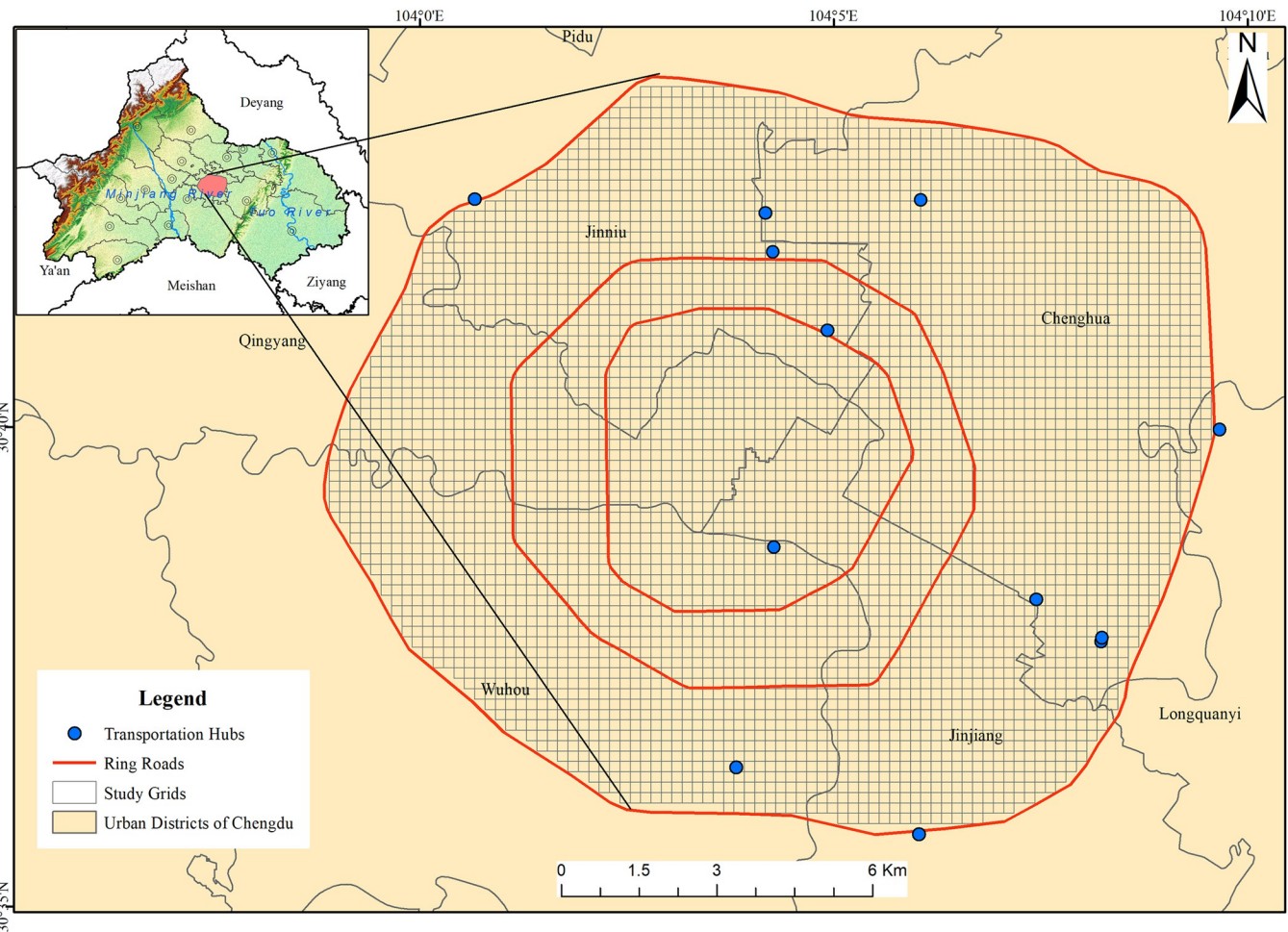

**Fig 1. Study area within third ring road in Chengdu and grid division.** (https://www.openstreetmap.org/copyright).

captured fairly well by using a 200m x 200m grid, with 4662 cells per square meter, to eliminate the effects of noisy data.

## 3.2. Data source

**3.2.1. Online car-hailing trips data.** The main data sets in this paper are the vehicle order trajectories, POIs, and other explanatory variables. We used the GAIA open data initiative (https://outreach.didichuxing.com/research/opendata/) from DiDi Chuxing to collect vehicle order trajectory data. These data are based on the DiDi big data advantage, which provides real desensitized trajectory data for academia. We selected GPS order trajectory data for 5 days between November 7 (Monday) and 11 (Friday), 2016, comprising a total of 1763480 records with an average of 352,696 trips per day. Redundant and out-of-domain data were removed. In each order track record, a unique identifier (order number), order start latitude, order start longitude, order end latitude, order end longitude, order start and end time were included.

Fig 2 shows the changes and average values of hourly DiDi trips during weekdays for one week. The three peak hours are at 9:00 a.m., 2:00 p.m. and 5:00 p.m. This indicates that there is more pressure on car use in the morning peak hours, evening peak hours and lunchtime, with a surge in demand for car-hailing. During the late night hours, as most of the residents are

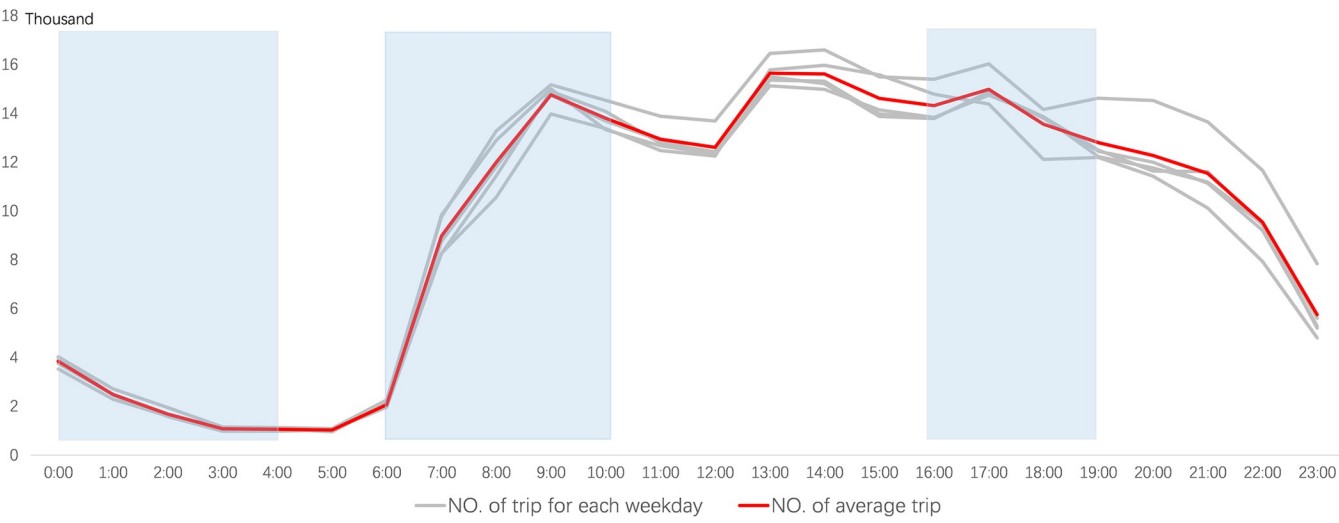

**Fig 2. Number of trips per hour on each weekday.**

sleeping, the number of DiDi trips reaches its lowest value of the day at around 4:00 a.m.. Because DiDi provides an all-day service while other transportation options cease during late night hours, and there is a smaller but essential level of demand from residents during the late night. In order to better reflect the spatial and temporal characteristics of residents' trips, this paper selects the morning and evening peak hours, when the traffic system is under the most pressure, and the late night hours, when it is more dependent on the night. These are morning peak hours (6:00 to 10:00), evening peak hours (16:00 to 19:00) and late night hours (0:00 to 4:00).

**3.2.2. Built environment variables.** The selection of independent variables is based on the "5D" principle proposed by Cervero and Ewing [10, 12]. As already noted above, the following variables were selected as potentially relevant for DiDi trips: number of POIs, internal transportation availability, road network density, population density, proximity to external transportation hubs, housing price, and diversity of regional facilities.

This paper quantifies the local road density in square kilometers. The data were obtained from the road network statistics of OpenStreetMap (https://www.openstreetmap.org/). The development of urban roads can be measured by this indicator, as well as by the spatial structure and size of neighborhoods at different stages of development in Chinese cities. This is an objective basis which describes the roads on which urban public transportation can operate smoothly. It is shown that taxis tend to be more willing to pass through areas where the density of the road network is higher [40].

The minimum official population statistics for Chinese cities are at the street and township level, small-scale population statistics are not available. The population data were derived from the WorldPop project (https://www.worldpop.org/) at the University of Southampton, UK, which were calibrated by the United Nations to generate 100 m resolution raster population size data for individual countries from 2000–2020. These high spatial resolution population data are widely used in many fields [41, 42].

In addition to car-hailing trips in cities, there are also public transportation methods such as subways, buses, and trains, which jointly influence residents' trips. Each individual transportation mode influences the others. To accurately explore the influence of internal and external transportation modes on car-hailing trips, the paper divided the POI transportation

facilities category into two categories: internal transportation availability and external transportation hubs. The number of bus stops and subway stations is taken as a measure of internal transportation facilities. There were 1674 internal transportation facilities in the Chengdu study area in November 2016.

Railway stations and bus stations in the study area were classified as external transportation hubs, and these intercity transportation facilities often have a large passenger flow. There were three railroad passenger stations and eight intercity bus stations in the Chengdu study area during the corresponding time period. Since these transportation facilities cover a large area and are generally connected by multiple entrances and roads, using the number of external transportation hubs as the only explanatory variable does not fully represent the hub characteristics. Therefore, a central coordinate was assigned to each intercity transportation facility, and a 200m buffer zone was established around this coordinate. The proportion of buffer zone area within a grid was used as the external transportation hub variable.

In most cities in China, there is no quantitative micro-indicator to measure regional or household income information. Based on regional average housing prices, we used household income data to examine how household income influences online car-hailing services [43]. In China, there is a significant correlation between regional house prices and household income. In the study, the average house price data for 9,100 neighborhoods in Chengdu in 2016 were obtained from the well-known Chinese real estate agency website Lianjia (https://cd.lianjia.com/). After comparing the results of multiple interpolation methods, kriging was found to be the most accurate and was selected to generate the average housing price raster data in the study area (Fig 3).

To measure the diversity of regional facilities, the Herfindahl-Hirschman Index (HHI) is introduced. The HHI is a comprehensive evaluation index reflecting the concentration of industries in a market and is commonly used in economics and government regulation [44]. A larger HHI indicates a higher concentration of industries and a greater monopoly of a particular industry. In this paper, the HHI was used to describe the diversity of regional facilities. A larger HHI indicates a higher regional concentration and more homogeneous facilities; a smaller HHI indicates a higher regional mixture and more diverse facilities. The HHI formula is as follows:

$$HHI_i = \sum_{j=1}^{n} \left( \frac{X_{ij}}{X_i} \right)^2 \tag{1}$$

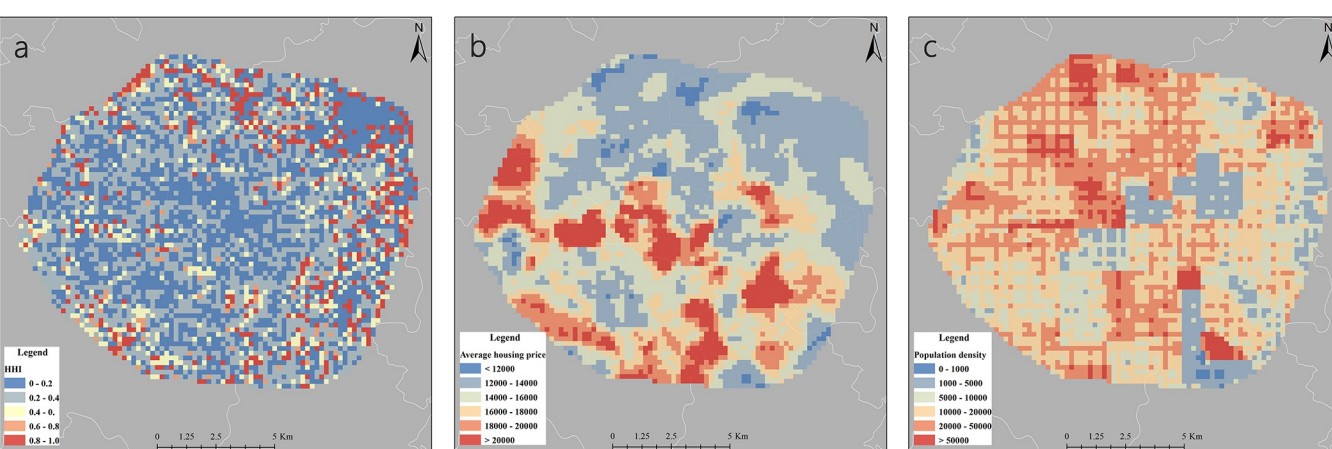

**Fig 3. The typical distributions of HHI, average housing price and population density.** a: HHI; b: average housing price; c: population density. (https://www.openstreetmap.org/copyright).

**Table 1. Built environment variables.**

| Variables | Description | Min | Mean | Max | Related Time |
|---|---|---|---|---|---|
| **Dependent Variables** | | | | | |
| Morning peak 1 | No. of pick-ups in morning peak hours | 0 | 5.62 | 97.0 | 6:00–10:00 |
| Morning peak 2 | No. of drop-offs in morning peak hours | 0 | 4.94 | 93.0 | 6:00–10:00 |
| Evening peak 1 | No. of pick-ups in evening peak hours | 0 | 6.60 | 183.8 | 16:00–19:00 |
| Evening peak 2 | No. of drop-offs in evening peak hours | 0 | 6.69 | 190.6 | 16:00–19:00 |
| Late night 1 | No. of pick-ups in late night hours | 0 | 1.57 | 69.8 | 0:00–4:00 |
| Late night 2 | No. of drop-offs in late night hours | 0 | 1.53 | 64.0 | 0:00–4:00 |
| **Independent Variables** | | | | | |
| **(1) POIs** | | | | | |
| Restaurant | No. of restaurants, cafes, etc. | 0 | 9.29 | 181 | 6:00–23:00 |
| Shopping | No. of shopping centers, retails, etc. | 0 | 13.53 | 507 | 8:00–22:00 |
| Living service | No. of post offices, barbershops, etc. | 0 | 10.63 | 238 | 8:00–22:00 |
| Corporation | No. of companies, enterprises, etc. | 0 | 2.95 | 136 | 8:00–22:00 |
| Financial service | No. of banks, securities companies, etc. | 0 | 0.62 | 19 | 8:00–23:00 |
| Education & culture | No. of schools, art galleries, etc. | 0 | 2.24 | 43 | 8:00–22:00 |
| Business residence | No. of residential communities, office buildings, etc. | 0 | 2.25 | 26 | 0:00–23:00 |
| Entertainment | No. of KTVs, bars, etc. | 0 | 1.34 | 74 | 0:00–23:00 |
| Medical facility | No. of hospitals, clinics, etc. | 0 | 1.94 | 37 | 0:00–23:00 |
| Governmental agency | No.of administrative agencies, police stations, etc. | 0 | 1.43 | 36 | 8:00–18:00 |
| Scenic spot | No. of parks, places of interest, etc. | 0 | 0.17 | 22 | 8:00–18:00 |
| Accommodation | No. of hotels | 0 | 2.24 | 249 | 0:00–23:00 |
| **(2) Internal transportation availability** | | | | | |
| Internal traffic | No. of bus stops and subway stations | 0 | 0.36 | 4 | 5:00–23:00 |
| **(3) Road network density** | | | | | |
| Road | Local road network density (km/km$^2$) | 0 | 9.70 | 72.42 | |
| **(4) Population density** | | | | | |
| Population | Population density(residents/km$^2$) | 675 | 20516 | 403250 | |
| **(5) Proximity to external transportation hubs** | | | | | |
| Transportation hub | Covered by train/intercity bus stations | 0 | 0.005 | 1 | 0:00–23:00 |
| **(6) Housing price** | | | | | |
| Housing price | Average houses price in the area | 10681 | 15788 | 36388 | |
| **(7) Diversity of regional facilities** | | | | | |
| HHI | Hirschman Herfindahl indices | 0 | 0.31 | 1 | |

where $X_{ij}$ represents the counts of facilities of category $j$ in the $ith$ region, and $X_i$ is the number of facilities of all categories in the $ith$ region. Fig 3 shows the distributions of HHI, average housing price and population density.

12 categories of POIs were selected from the Gaode Map (https://lbs.amap.com) data set, including restaurants, living services, business residences and other variables. Table 1 shows all the variables used. For ease of expression, some abbreviated variables are used in the following.

## 3.3. Methods

Unlike the classical GWR model, MGWR uses different bandwidths for each variable and requires different levels of spatial smoothing for each variable; therefore, the model results can better represent the observations, yielding a smaller fitting error. The MGWR formula is as

follows [25, 34]:

$$y_m = \beta_0(u_m, v_m) + \sum_{i=1}^{k} \beta_{bwi}(u_m, v_m)x_{mi} + \varepsilon_m \qquad (2)$$

where $y_m$ is the attribute value at position $m$; a regression coefficient's bandwidth is defined as $bwi$ for the $ith$ variable; $\beta_0(u_m, v_m)$ is the center coordinate at position $m$; $\beta_{bwi}(u_m, v_m)$ is the regression coefficient of the $ith$ variable at $m$; and $\beta_0(u_m, v_m)$ and $\varepsilon_m$ are the intercept and error term of the model at position $m$, respectively.

Based on local regression, the MGWR regression coefficient $\beta_{bwi}$ is derived–this part is the same as that of the classical GWR model. Then, using a Gaussian kernel function and corrected Akaike Information Criterion (AICc), the MGWR model complies with the GWR model's bandwidth selection and kernel function. The MGWR's bandwidth heterogeneity is the key difference to the GWR, and can be implemented as generalized additive models (GAMs) [45]. GAMs use a backward-fitting algorithm to fit each smoothing term: all smoothing terms are first set up initially, the model then calculates the difference between the true and predicted values using the classic GWR as the initial estimate, and the initialized residual $\hat{\varepsilon}$ is obtained. The formula is as follows [45]:

$$\hat{\varepsilon} = y - \sum_{i=1}^{k} \hat{f}_i \quad (f_i = \beta_{bwi}x_i) \qquad (3)$$

A classical regression with GWR is conducted by regressing the residuals $\hat{\varepsilon}$ plus the first additive term $\hat{f}_1$, to replace the previous parameter estimates with a new column of bandwidth estimates $bw1$, a new column of parameter estimates $\hat{f}_1$ and $\hat{\varepsilon}$ must be computed. This step is repeated to the last variable, and the residuals plus the $ith$ additive term $\hat{f}_i$ are regressed against the $(i-1)th$ independent variable. Then the parameter estimates $\hat{f}_i$ and $\hat{\varepsilon}$ are updated for the $ith$ variable. The above integral step is repeated until a convergence criterion is satisfied. This paper uses the classical residual sum of squares (RSS) as a convergence criterion:

$$SOC_{RSS} = \left| \frac{RSS_{new} - RSS_{old}}{RSS_{new}} \right| \qquad (4)$$

where $RSS_{old}$ represents the RSS in the previous step, $RSS_{new}$ represents the RSS in this step.

The calculations of the MGWR model in this study are based on the Spatial Analysis Research Center (SPARC) at Arizona State University, USA. The MGWR2.2 software is developed by SPARC (https://sgsup.asu.edu/SPARC), the maps are produced using ArcGIS 10.6 software.

## 4. Results

S1 and S2 Tables show the results of Pearson correlation analysis, collinearity diagnostics, and global Moran index test. Most of the variables selected for this study demonstrate some independence and spatial autocorrelation, and there are no strongly collinear variables, indicating that the data are suitable for subsequent modeling in this paper. Specifically, in the correlation test, the correlations with transportation hub are insignificant, and the other variables show mild correlation with correlation coefficients below 0.8. In the collinearity diagnostics, all variables have variance inflation factors (VIF) less than 10; all variables in the spatial autocorrelation test have significant positive Z scores with p-values close to 0, highlighting the obvious spatial clustering characteristics. In summary, the transportation hub variables with insignificant correlation are excluded from this paper.

As shown in Table 2, MGWR achieves a better $R^2$ than GWR and OLS, and a lower AICc than that for the classical GWR, for all study periods. The highest $R^2$ is 0.652 for the evening

**Table 2. Comparison between global and local models.**

| Metrics | Morning peak 1 model (pick-ups) | | | Morning peak 2 model (drop-offs) | | | Evening peak 1 model (pick-ups) | | |
|---|---|---|---|---|---|---|---|---|---|
| | OLS | GWR | MGWR | OLS | GWR | MGWR | OLS | GWR | MGWR |
| AICc | 12785.6 | 11243.7 | 10824.0 | 12799.3 | 11313.6 | 10815.5 | 12645.9 | 11252.1 | 10197.3 |
| ENP | | 511.3 | 587.2 | | 515.0 | 586.6 | | 595.0 | 774.0 |
| RSS | 4225.2 | 2378.3 | 2084.3 | 4235.8 | 2409.4 | 2081.3 | 4095.2 | 2274.8 | 1632.3 |
| $R^2$ | 0.094 | 0.490 | 0.553 | 0.090 | 0.483 | 0.554 | 0.122 | 0.512 | 0.650 |
| Metrics | Evening peak 2 model (drop-offs) | | | Late night 1 model (pick-ups) | | | Late night 2 model (drop-offs) | | |
| | OLS | GWR | MGWR | OLS | GWR | MGWR | OLS | GWR | MGWR |
| AICc | 12649.9 | 11248.2 | 10198.0 | 12854.9 | 12238.6 | 11802.2 | 12851.6 | 12228.0 | 11794.1 |
| ENP | | 602.6 | 784.8 | | 312.5 | 548.8 | | 312.6 | 546.2 |
| RSS | 4100.5 | 2263.2 | 1621.7 | 4292.2 | 3262.6 | 2626.6 | 4289.2 | 3255 | 2625.7 |
| $R^2$ | 0.120 | 0.515 | 0.652 | 0.080 | 0.300 | 0.437 | 0.080 | 0.302 | 0.437 |

Note: AICc: Akaike information criterion; ENP: Effective number of parameters; RSS: Residual sum of squares.

peak 2 model, with fewer effective data and a lower RSS. The MGWR model provides regression-based estimates that are closer to the observations, by using fewer parameters. At the same time, it can directly reflect the differences in action scales between different variables. Overall, the MGWR model outperforms the GWR and OLS models in all aspects.

## 4.1. Spatio-temporal characteristics of residential online car-hailing trips

Fig 4 shows the spatial distributions of DiDi trips during the three periods of a weekday, with a clear spatial and temporal divergence. Specifically, among the three time periods in the study area, the total number of late night DiDi trips is the lowest, and the distributions of pick-up and drop-off locations is basically the same, mainly in the vicinity of major commercial centres in the north, east and south of the Second Ring Road, such as Chunxi Road, Renmin North Road and Jianshe North Road. This is a time of less travel for residents, most of whom are already resting, and the main commercial centres within the Second Ring Road have a higher flow of people due to the presence of a thriving night market commercial culture; the Chengdu Railway Station, which is a 24-hour operation for high-speed trains and trains, still has trains arriving late at night, and the intensity of travel is still high at this stage; young university students are one of the main groups of users of DiDi, and the heat of pick-up and drop-off is higher in the vicinity of universities late at night.

During the morning peak, the number of DiDi trips increases substantially compared to the late night, and the distributions of travel hotspot areas is centered on the commercial centre areas within the second ring, with a sparse distributions in a radial pattern towards the residential districts in the third ring area, while the distributions of drop-off points is more closely concentrated than the regional distributions of pick-up points, with the obvious commuting characteristics of the inflow from the residential areas in the outer ring to the central commercial work areas, which is consistent with previous studies [46].

However, in the evening peak, we find that this commuting characteristic does not have obvious reverse outflow phenomenon in the night, although there is the most travel volume and travel hot area in this period among all the study periods. This may be due to the fact that after-work residents do not have a significantly more stressful commute than when they go to work. On the one hand, after-work residents tend to have more time to choose other more economical modes of public transport rather than the more expensive option of an online car-hailing; on the other hand, in addition to the general need to commute home, many residents

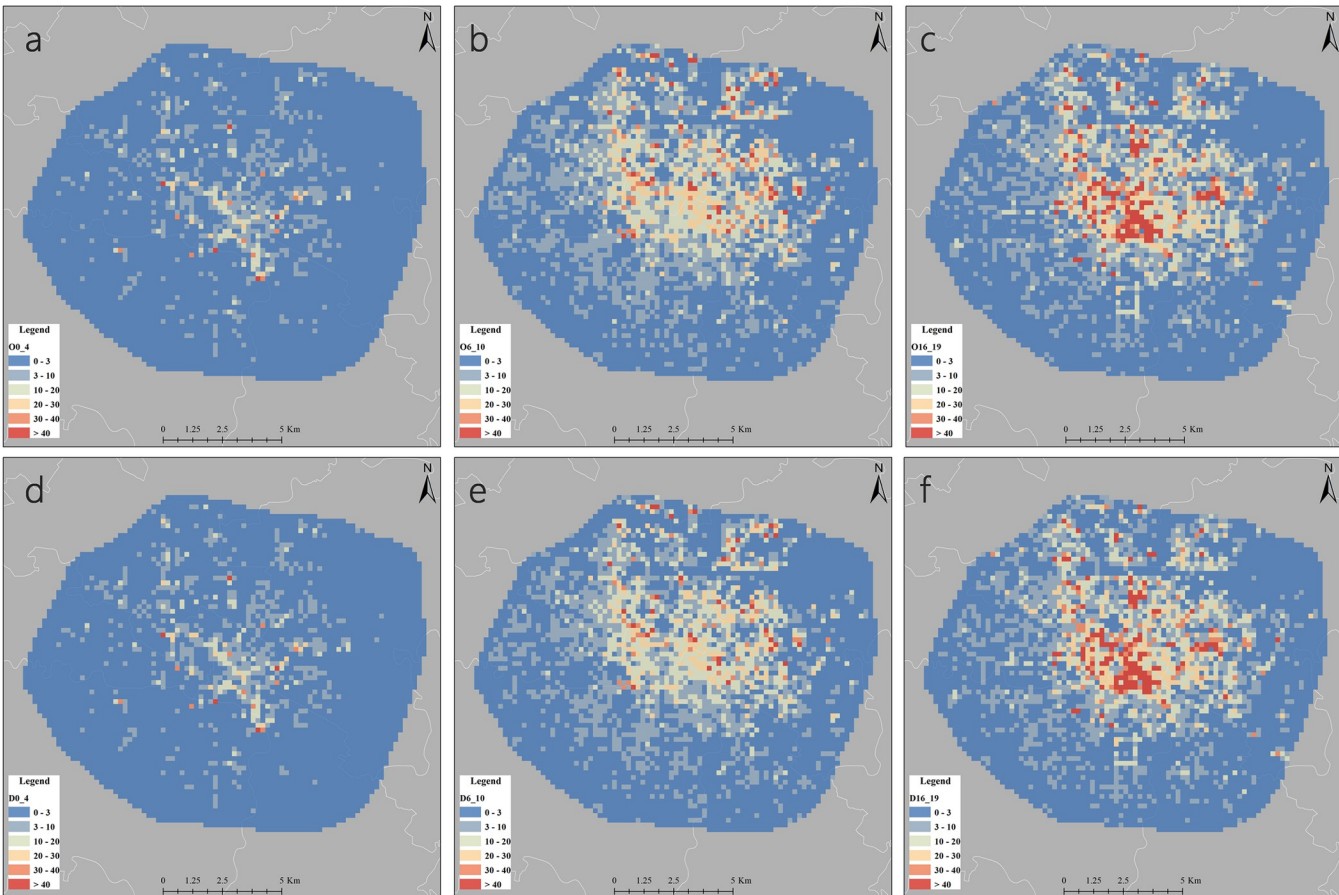

**Fig 4. A typical weekday's average number of pick-ups and drop-offs.** a/d: Morning peak 1/Morning peak 2 b/e: Evening peak 1/Evening peak 2 c/f: Late night 1/Late night 2. (https://www.openstreetmap.org/copyright).

do not have a significant outflow from the central business district as they have a need for entertainment and leisure near the commercial centre. The bandwidths of the business residence, governmental agency, population and housing price variables are almost identical in the two models at this stage, which is also reflected in the standard deviations of the coefficients of the variables.

## 4.2. Results of multi-scale geographically weighted regression

Table 3 shows the regression results of the MGWR model and the different pick-up and drop-off variables during the morning peak. The table mainly shows the localized spatial variation of the independent variables, as indicated by relevant statistical metrics such as mean, minimum and maximum values. In the morning peak, the pick-up location passes the model significance test for business residence, living service, governmental agency, population, and housing price. The corporation passes the test and is positively correlated in the drop-off model, illustrating the significant commuting characteristics of residents' DiDi trips in the morning peak 1 model, which are strongly influenced by the destination of work. However, the bandwidth of living service and constant variables is obviously different from other variables, and the elasticity of coefficient values in the drop-off model is slightly larger than that in the pick-up model. This indicates that the influence scope of business residence, governmental

**Table 3. Pick-ups and drop-offs during the morning peak.**

Morning peak 1 model (pick-ups)

| | Mean | Std | Min | Median | Max | Bandwidth | VIF |
|---|---|---|---|---|---|---|---|
| Intercept | 0.003 | 0.628 | -0.631 | -0.244 | 2.511 | 230.00 | |
| Business residence | 0.032 | 0.000 | 0.031 | 0.032 | 0.032 | 34789.39 | 1.414 |
| Living service | 0.020 | 0.039 | -0.025 | 0.006 | 0.129 | 2527.12 | 1.243 |
| Governmental agency | -0.013 | 0.000 | -0.013 | -0.013 | -0.013 | 34789.93 | 1.179 |
| Population | 0.001 | 0.000 | 0.001 | 0.001 | 0.001 | 34790.26 | 1.007 |
| Housing price | 0.007 | 0.000 | 0.007 | 0.007 | 0.008 | 34790.26 | 1.003 |

Morning peak 2 model (drop-offs)

| | Mean | Std | Min | Median | Max | Bandwidth | VIF |
|---|---|---|---|---|---|---|---|
| Intercept | 0.002 | 0.630 | -0.619 | -0.249 | 2.512 | 230.43 | |
| Business residence | 0.033 | 0.000 | 0.033 | 0.033 | 0.033 | 34789.39 | 1.462 |
| Living service | 0.024 | 0.041 | -0.022 | 0.009 | 0.136 | 2461.41 | 1.253 |
| Governmental agency | -0.008 | 0.000 | -0.008 | -0.008 | -0.008 | 34789.93 | 1.197 |
| Population | 0.001 | 0.000 | 0.001 | 0.001 | 0.001 | 34790.26 | 1.007 |
| Housing price | 0.007 | 0.000 | 0.007 | 0.007 | 0.007 | 34790.26 | 1.003 |
| Corporation | 0.029 | 0.008 | 0.019 | 0.068 | 0.107 | 5120.20 | 1.124 |

agency, population and housing price variables in different research grids is nearly the same, that is, they do not show obvious spatial heterogeneity, while the spatial heterogeneity of life and constant variables is obvious.

The regression findings for the evening peak are shown in Table 4. Due to the increased number of DiDi trips during the evening peak, more variables passes the model test and has the highest model fit among the three models, with R-squared of 0.65 and 0.652 for the drop-off and pick-up models respectively, which achieves better model results compared to other scholars. The mean coefficients of the pick-up and drop-off variables are similar, with living service, restaurant, financial service, internal traffic, and accommodation all positively correlated, and business residence, governmental agency, housing price, and scenic spot negatively correlated. The bandwidths of housing price, restaurant, financial service, scenic spot, internal traffic, and accommodation are similar, such that the variable regression coefficients take fixed values, with standard deviations of 0. The bandwidths of the business residence, living service, governmental agency and constant vary significantly, and their coefficients vary widely, but their standard deviations all fluctuate around zero, and they have relatively stable coefficient elasticity. Compared to the morning peak model, the number of DiDi trips is higher and the model variables tested are more diverse as residents have more free time and more diverse travel demand after work, which confirms that there is no significant commuting outflow in the spatial and temporal distributions of trips during the evening peak.

The late night model's regression results are shown in Table 5. In this time period, the total number of DiDi trips is the smallest. Since most of the facilities related to explanatory variables are closed at night, in order to increase the authenticity and reliability of the model, this paper eliminates shopping, living service, corporation, financial service, education & culture, governmental agency, scenic spot, and internal traffic, and uses the remaining 24-hour variables or objective variables for the regression. Finally, the business residence, entertainment, and accommodation variables show significant correlations with DiDi trips, and according to the goodness of fit, these three variables can explain 43.7% of pick-ups and drop-offs during the late night, while medical facility, road, population and HHI do not pass the model test. Among

**Table 4. Pick-ups and drop-offs during the evening peak.**

Evening peak 1 model (pick-ups)

|  | Mean | Std | Min | Median | Max | Bandwidth | VIF |
|---|---|---|---|---|---|---|---|
| Intercept | 0.015 | 0.716 | -0.640 | -0.232 | 6.829 | 204.91 | |
| Business residence | 0.029 | 0.002 | 0.026 | 0.029 | 0.033 | 9335.38 | 1.802 |
| Living service | -0.023 | 0.031 | -0.082 | -0.024 | 0.059 | 2192.23 | 1.592 |
| Governmental agency | 0.022 | 0.135 | -0.236 | -0.001 | 0.984 | 736.73 | 1.190 |
| Housing price | 0.041 | 0.000 | 0.041 | 0.041 | 0.041 | 34790.26 | 1.029 |
| Restaurant | -0.027 | 0.000 | -0.027 | -0.027 | -0.026 | 34789.39 | 1.891 |
| Financial service | -0.008 | 0.000 | -0.008 | -0.008 | -0.008 | 34788.51 | 1.226 |
| Scenic spot | 0.001 | 0.000 | 0.001 | 0.001 | 0.002 | 34789.39 | 1.017 |
| Internal traffic | -0.024 | 0.000 | -0.024 | -0.024 | -0.024 | 34790.26 | 1.065 |
| Accommodation | -0.004 | 0.000 | -0.004 | -0.004 | -0.004 | 34789.93 | 1.122 |

Evening peak 2 model (drop-offs)

|  | Mean | Std | Min | Median | Max | Bandwidth | VIF |
|---|---|---|---|---|---|---|---|
| Intercept | 0.016 | 0.715 | -0.653 | -0.228 | 6.954 | 204.12 | |
| Business residence | 0.029 | 0.002 | 0.025 | 0.029 | 0.033 | 8894.73 | 1.802 |
| Living service | -0.019 | 0.024 | -0.060 | -0.021 | 0.046 | 2464.25 | 1.592 |
| Governmental agency | 0.025 | 0.153 | -0.256 | -0.002 | 1.214 | 674.11 | 1.190 |
| Housing price | 0.044 | 0.000 | 0.044 | 0.044 | 0.044 | 34790.26 | 1.029 |
| Restaurant | -0.030 | 0.000 | -0.030 | -0.030 | -0.030 | 34789.39 | 1.891 |
| Financial service | -0.011 | 0.000 | -0.011 | -0.011 | -0.011 | 34781.09 | 1.226 |
| Scenic spot | 0.003 | 0.000 | 0.003 | 0.003 | 0.003 | 34789.39 | 1.017 |
| Internal traffic | -0.023 | 0.000 | -0.023 | -0.023 | -0.023 | 34790.26 | 1.065 |
| Accommodation | -0.004 | 0.000 | -0.004 | -0.004 | -0.004 | 34789.93 | 1.122 |

them, entertainment, business residence and accommodation are positively correlated with taxi trips. These variables exhibit significant bandwidth differences, indicating that the spatial heterogeneity of the explanatory variables is stronger during the late night. This also shows that there is more demand for entertainment and accommodation during the late night, in addition to the need to get home. In late night, although the total number of taxi trips is less, the regression results of this model are still of strong practical significance because residents are more dependent on online car-hailing or taxis in late night.

**Table 5. Pick-ups and drop-offs during the late night.**

Late night 1 model (pick-ups)

|  | Mean | Std | Min | Median | Max | Bandwidth | VIF |
|---|---|---|---|---|---|---|---|
| Intercept | -0.003 | 0.534 | -0.480 | -0.193 | 3.378 | 238.05 | |
| Business residence | 0.013 | 0.027 | 0.006 | 0.015 | 0.058 | 34789.39 | 1.142 |
| Entertainment | 0.060 | 0.035 | 0.011 | 0.069 | 0.214 | 2712.59 | 1.110 |
| Accommodation | 0.062 | 0.025 | 0.005 | 0.071 | 0.090 | 3153.25 | 1.118 |

Late night 2 model (drop-offs)

|  | Mean | Std | Min | Median | Max | Bandwidth | VIF |
|---|---|---|---|---|---|---|---|
| Intercept | -0.003 | 0.535 | -0.478 | -0.195 | 3.246 | 238.59 | |
| Business residence | 0.013 | 0.026 | 0.003 | 0.016 | 0.065 | 34789.39 | 1.142 |
| Entertainment | 0.059 | 0.034 | 0.018 | 0.069 | 0.002 | 2775.45 | 1.110 |
| Accommodation | 0.062 | 0.025 | 0.005 | 0.071 | 0.090 | 3159.25 | 1.118 |

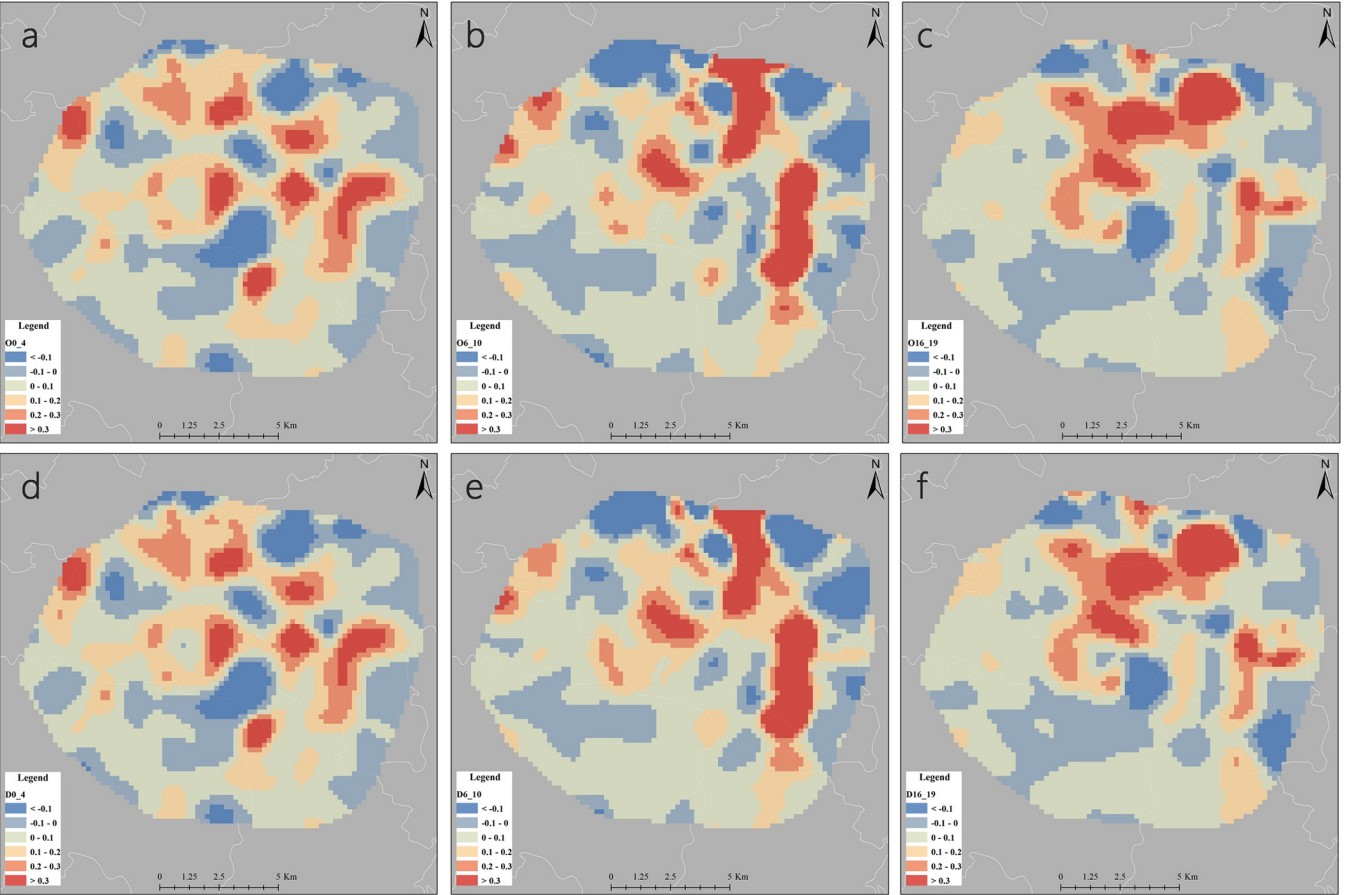

**Fig 5. Coefficients of business residence's spatial-temporal distributions.** a/d: Morning peak 1/Morning peak 2 b/e: Evening peak 1/Evening peak 2 c/f: Late night 1/Late night 2. (https://www.openstreetmap.org/copyright).

## 4.3. Coefficients of business residence's spatial-temporal distributions

The business residence variable passes all the tests of the model and we conduct a cartographic analysis of the spatio-temporal distributions of the coefficients of the business residence variable at different time periods, as shown in Fig 5. With the exception of the southern part of the central city and some areas on the edge of the Third Ring, most areas have positive model estimates during the study period, indicating that the residential variable is significantly and positively associated with residential DiDi trips, and consistent with previous related studies, residential factors have a significant impact on car-hailing trips [47, 48]. Among the morning peak 1, morning peak 2 and evening peak 1 models, the spatial distributions of business residence variables show some similarity due to the obvious commuting characteristics. In the evening peak 2 model, the spatial distributions of business residence coefficients differs from that of the other models, due to the diverse travel demand of residents at night. The late night model captures the residents' need for leisure time, leading to high absolute values in major commercial areas and in some residential neighborhoods.

## 5. Discussion

It has been demonstrated that urban built environment factors such as density, diversity, and accessibility have significant effects on residents' travel behavior [10, 12]. In this study, the

relationship between built environment characteristics and online car-hailing pick-up and drop-off in Chengdu, China, was investigated. The global OLS model, local GWR model and MGWR model each confirm that the selected data on drop-off and pick-up of DiDi trips are significantly correlated with built environment variables. The goodness of fit of the morning peak and evening peak models is significantly improved when compared with previous studies at the same scale [11, 24]. Overall, in this paper, the regression results of the MGWR model are better than those of the GWR and OLS models, allowing a better reflection of spatial heterogeneity in the roles of different variables. Some key variables in the model are also explored.

Except for two models for late night, the correlation coefficients of the housing price variable in the remaining four models that pass the significance test are all positive, indicating that there is a positive correlation between housing price and DiDi trips. As a representative of residents' income level, to a certain extent, the housing price can reflect the consumption level of residents' households. On the one hand, the higher the income level of residents, the more likely they are to take a taxi compared to other transportation modes. On the other hand, areas with higher housing price are located in the core business districts of Chengdu, such as Cao Tang Street, Chunxi Road, and Dongda Road, which are prone to serious traffic congestion caused by intensive traffic flow in the morning peak and evening peak, encouraging commuting by taxi instead of driving. This result supports similar findings in previous studies [11, 49]. In contrast, during the late night, it is difficult for the housing price variable to explain late-night car-hailing trips, due to the lower total counts of DiDi trips and the need for diverse leisure needs of residents and a small number of overtime commuters.

The coefficients of the business residence variable are higher during morning peak and evening peak than at late night, and residential trips are more likely to be influenced due to the needs of commuters. During the late night, DiDi trips are influenced by a combination of business residence, entertainment, and accommodation variables, perhaps reflecting the demand of residents for leisure and a small number of overtime commuters. In terms of spatial distributions, low coefficients are mainly located in the eastern and northern areas at the edge of the ring road. In comparison, the northern, northeastern, and eastern regions have lower housing price (Fig 3). According to the survey, this part of the region is mainly in Jinniu District and Chenghua District, and is dominated by warehousing, logistics and manufacturing plants. Consequently, foreign laborers, wage earners and low-income families tend to live in this area. On the other hand, in the southern and southwestern regions, the population is lower but has a higher level of housing price, and its business residence coefficient fluctuates around zero. This is mainly an affluent residential area in the traditional sense, where convenient private cars make residents more likely to travel by car, but higher income levels also allow them to accept the burden of traveling by taxi. Overall, they are likely to have a higher degree of flexibility for DiDi trips.

The correlation coefficient for the internal traffic variable is negative in the evening peak model, which means that residents' choice for DiDi trips when commuting from work is still strongly influenced by the subway and public transportation. Residents may be more willing to take the more cost-effective public transportation when in close proximity to subway and bus stations, all other things being equal. Meanwhile, residents located farther away from public transportation services will have less access to them, and are more likely to take a taxi for their off-duty commute. In addition, more planned public transportation trips by residents during the morning peak may cause the internal traffic variable to have a reduced effect on DiDi trips, such that the internal traffic does not pass the significance test during this period.

Among the other land use type variables, corporation is significantly positively correlated in the morning peak 2 model but insignificantly correlated in the evening peak model, probably due to more diverse travel demand and destinations. Scenic spot is positively correlated in

the evening peak model, which indicates that the proximity to scenic spot is also one of the potential sources of DiDi trips. During the late night, in addition to the demand for entertainment, the accommodation demand also has a significant impact on DiDi trips. Unexpectedly, the road, HHI index and intercity transportation facilities variables do not pass the model significance test, which may be related to the small-scale research grid employed in this paper.

## 6. Conclusion

This paper explored the association between online car-hailing and built environment characteristics in Chengdu, at the small grid scale, based on the MGWR model. The main findings are summarized as follows. First, we set up a 200m small-scale research grid, and analyse the spatial and temporal distributions characteristics of residents' online car-hailing trips using DiDi order trajectory data, and extract the typical time periods as our research periods. Second, after a comparative study under this grid, the MGWR model fits better than the GWR and OLS models in this paper, and some of the models fit better than related studies by previous scholars. Our study demonstrates the good applicability of the MGWR model when studying the relationship between DiDi trips and built environment in small-scale study area. The MGWR model shows that housing price, population, and internal traffic have a significant effect on residents' trips in the peak hour model; business residence is positively associated with pick-up and drop-off in all models; and entertainment, scenic spot, and accommodation have a stronger effect on DiDi trips in evening peak and late night. Finally, the study finds significant spatial heterogeneity in the impact of built environment variables on car-hailing trips, of which we explore the spatial heterogeneity of variables for business residence that passes all model tests. The coefficients of the business residence variable are found to vary significantly across regions due to a number of factors including population, average housing price, population density and industrial structure layout. These findings can be used to inform the relevant authorities in some of the following ways:

First, during weekdays, there are distinct peak and low hours for residents' car-hailing, with residents having higher demand for travel between 6:00–10:00, 12:00–14:00 and 16:00–19:00 than during other hours. When adjusting their operating strategies, transport authorities should focus on these peak hours and should adjust or reserve vehicle capacity to cope with them. At the same time, with limited vehicles and road infrastructure, they should encourage carpooling by residents during peak hours to reduce the pressure on car-hailing trips.

Second, in the spatial distributions of residents' trips throughout the day, the volume and extent of trips during the late night are the smallest, mainly concentrated in the major commercial centres in the north, east and south areas of the Second Ring Road, near railway stations and university campuses; during the morning and evening peak, the hot areas for trips are centred on major commercial areas such as Chunxi Road and Jianjian North Road within the Second Ring Road, and are sparsely distributed in a radial pattern along the main roads and residential areas towards the Third Ring Road. The maximum number of trips per day occurs during the evening peak. Online car-hailing companies should allocate appropriate vehicles to these hot areas of demand in advance to reduce unnecessary waiting for drivers and passengers. Residents should also try to stagger their trips during peak hours to avoid areas of peak demand for taxis, which is more conducive to convenient and efficient travel.

Third, the business residential variable has a significant impact on the amount of residential trips. Housing price, population and internal traffic have a significant impact on residential trips during peak hours, and residents have a greater demand for leisure and entertainment during the evening peak and late night. When cruising, online car-hailing drivers can precisely travel to areas with more of these variables to find passengers and tend to have a greater chance

of picking up passengers. Also, planning the location of future journeys in advance can help drivers effectively avoid traffic congestion and improve the efficiency of urban mobility.

Fourth, the built environment variables affecting residential trips are significantly spatially heterogeneous. During peak hours, the business residence variable has a significant negative effect on DiDi trips in the edge of the Third Ring Road and the southern part of the Second Ring Road, and a significant positive effect on DiDi trips in the northern, eastern, southeastern and northern edge of the Second Ring Road, indicating that the residential variable has a large spatial heterogeneity in its effect on residents' trips, which is mainly caused by the combination of factors such as regional population density, income level and industrial layout. The lack of supporting facilities can easily lead to uneven distributions of traffic resources and greater pressure on traffic, and a reasonable variety of supporting facilities is more conducive to convenient travel and efficient operation of urban traffic in the future.

Fifth, in general, residential demand for car-hailing trips is influenced to a large extent by variables in the built environment within a region and by the joint effects between neighbouring regions. Although a single variable in the built environment has an impact on residents' trips by DiDi, it does not explain the entire travel process of residents. Therefore, in the process of urban planning by urban authorities, the division of urban functional areas should be taken into account and the internal structure of urban spatial units should be further optimised.

## 7. Limitations and future work

There are still some limitations here that need to be addressed in the future.

First, in addition to the influence of the built environment, the age, gender and education quality of the residents also have a strong subjective effect on the online car-hailing trips. However, due to data acquisition constraints, the corresponding population data are not available in Chengdu, more research on subjective factors of residents is needed in the future.

Second, this paper adopts a small-scale research grid of 200m x 200m. Consequently, there may be schools, hospitals and other large units that are divided, resulting in statistical errors. In addition, the trip destination is not always in the same neighborhood as the drop-off, and there is the possibility of travel to adjacent grid cells depending on the influence of the surrounding POI. How to measure this influence is still pending further research.

Third, in terms of the study period and region, the paper only selects data representative of DiDi trips within Chengdu's third ring on weekdays. The dataset does not expand to other regions and time periods due to the more remote location of the airport and the atypical commuting characteristics on weekends. Future studies with more data and in a larger study area should be considered to further validate our findings.

## Supporting information

**S1 Table. Spatial autocorrelation.**
(DOCX)

**S2 Table. Correlation coefficient test table for each variable.**
(DOCX)

## Acknowledgments

We thank those anonymous reviewers and the editor whose comments/suggestions helped improve and clarify this manuscript.

## Author Contributions

**Conceptualization:** Yan Cao, Yongzhong Tian, Jinglian Tian.

**Methodology:** Yan Cao, Yongzhong Tian.

**Supervision:** Jinglian Tian, Kangning Liu, Yang Wang.

**Validation:** Yan Cao, Yongzhong Tian.

**Writing – original draft:** Yan Cao.

**Writing – review & editing:** Jinglian Tian, Kangning Liu, Yang Wang.

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
