## [Decision Letter · Decision Letter 0]

16 Aug 2022

PONE-D-22-20651Analysis of urban residents' online car-hailing journey characteristics based on the MGWR modelPLOS ONE

Dear Dr. Tian,

Thank you for submitting your manuscript to PLOS ONE. After careful consideration, we feel that it has merit but does not fully meet PLOS ONE’s publication criteria as it currently stands. Therefore, we invite you to submit a revised version of the manuscript that addresses the points raised during the review process.

Specifically, the reviewers recommend major revisions which include adding new and recent references, an explicit explanations of research findings, explanation of the novelty of research, language and grammar issues and others. You can find the details of reviewers' reports as attached to this e-mail.

We look forward to receiving your revised manuscript.

Kind regards,

Eda Ustaoglu, PhD

Academic Editor

PLOS ONE

Journal Requirements:

2. Please note that PLOS journals require authors to make all data necessary to replicate their study’s findings publicly available without restriction at the time of publication. Please see our Data Availability policy at https://journals.plos.org/plosone/s/data-availability. As such, please make the full specific dataset used in this study available by either A) uploading the full dataset as supplementary information files, or B) including a URL link in your Data Availability Statement and Methods section to where the full dataset can be accessed.

4. We note that Figures 1, 3, 4 and 5 in your submission contain [map/satellite] images which may be copyrighted. All PLOS content is published under the Creative Commons Attribution License (CC BY 4.0), which means that the manuscript, images, and Supporting Information files will be freely available online, and any third party is permitted to access, download, copy, distribute, and use these materials in any way, even commercially, with proper attribution. For these reasons, we cannot publish previously copyrighted maps or satellite images created using proprietary data, such as Google software (Google Maps, Street View, and Earth). For more information, see our copyright guidelines: http://journals.plos.org/plosone/s/licenses-and-copyright.

a. You may seek permission from the original copyright holder of Figures 1, 3, 4 and 5 to publish the content specifically under the CC BY 4.0 license.  

Reviewers' comments:

Reviewer's Responses to Questions

**Comments to the Author**

1. Is the manuscript technically sound, and do the data support the conclusions?

Reviewer #1: Partly

Reviewer #2: Yes

2. Has the statistical analysis been performed appropriately and rigorously? 

Reviewer #1: I Don't Know

Reviewer #2: No

3. Have the authors made all data underlying the findings in their manuscript fully available?

Reviewer #1: Yes

Reviewer #2: Yes

4. Is the manuscript presented in an intelligible fashion and written in standard English?

Reviewer #1: No

Reviewer #2: No

5. Review Comments to the Author

Reviewer #1: The paper improves the method and data to explore the issue of urban online car-hailing. It reflects a large amount of work, but still has distance from a qualified paper.

Major revisions:

1. The paper is poorly expressed in language. Pay attention to the mixed use of past and present tense, especially, in the same sentence.

2. There are too many abbreviations in the paper, which seriously affects the reading as well as the understanding of the contents.

3. The introduction needs to be reorganized to highlight the value and meaning of the research. Too much discussion of variables selection is not appropriate for the introduction.

4. The discussion of the results is mainly on the sign of the variables, and the analysis of the causes of the effects is not in-depth. The number of variables of interest are too much, resulting in unfocused findings.

5. The conclusion is brief and the policy recommendations lack relevance and application.

6. The title is not specific enough and does not indicate the core contents of the study. The introduction section points out "the purpose of this study is to examine the spatiotemporal relationship between online cab-hailing and the urban built environment", but the title is a study of "characteristics", which is confusing. It is recommended to revise it to cover and reflect your core research contents.

Other comments:

7. Unify the expressions of the core terminology in the whole paper.

(1) Such as: car-hailing or cab-hailing? Please explain to me the difference between the two concepts and why you chose this one and not the other? If the two concepts refer to the same content, please unify the expressions in the text directly.

(2) DiDi has an English version of its website, please check the English spelling of DiDi. Please unify the expressions in the whole text.

8. L67: What are the others referring to?

9. L141-L147:

(1) There is actually another peak(13:00-14:00) in Figure 2, why is this not discussed?.

(2) MP and EP? In the case of abbreviations for the first time, please use the whole words.

(3) The presentation of the content is illogical, and the English expression level is very poor.Please have your paper copy edited by a native speaker. Pay attention to the mixed use of past and present tense, especially, in the same sentence. Be consistent.

10. Section 2.2 contains too much contents and the logic is not clear enough. Please rearrange the logic of Section 2 and divide it into a proper chapters.

11. L229: (1)Where are the Tables A1 and A2? (2)Poor English proficiency and confusing use of tenses in this paragraph.

12. L237~L248: (1)Lack of standardization in table designing. (2)Please correspond to Table 1 for the order of variables in Table 2. (3)What is the difference between R-square and R2 in Table 2? (4)L241: EP? OR EPP? (5)It is not appropriate to discuss the conclusions of Table 4 here. (6)Please explain the meaning of AICc in the note part of Table 2. (7)Please explain on what software or algorithm these results are based?

13. Some of the details:

L124: I don't think it's appropriate for the contents to appear in this sentence: "rich and diverse internal land use variables (variables?)".

Line126: I don't think the contens of the brackets, (such as Jinniu and Chenghua), make sense.

L140, L147: A period is missing at the end of the sentence.

L159: [Please check that I have understood this section correctly.]?

L194: distributions

L197: Table1, the explained variables are usually placed before the explanatory variables.

The Table 4 and Table 5 are misplaced. When discussing tables, please place the table below the text.

Conclusion: Please note the use of punctuation in the whole test.

Reviewer #2: I am glad to review and assess this exciting article, entitled, "Analysis of urban residents' online car-hailing journey characteristics based on the MGWR model”. The organization of this article is good and satisfactory. The Introduction section, and methodology portions are adequate. I suggest the authors improve these parts overall to enhance the work quality. As suggested, I recommend that authors do a little more work and add the latest literature to support the study. I accept this manuscript after major revision, as I have recommended. Some valuable comments are given below. Please do a little work and improve the manuscript

1. The manuscript needs language, grammar, and syntactic editing. The English language usage should be checked by a fluent English speaker.

2. The more suitable title should be selected for the article

3. Also, find some new related references which should add to the 1st and 2nd parts.

4. Must add much more explanations and interpretations for the results, which are not enough. It is suggested to compare the results of the present research with some similar studies which is done before (more justification is needed).

5. Please make sure your conclusions section underscores the scientific value-added of your paper and the applicability of your findings/results, as indicated previously. Please revise your conclusion part into more detail. It would be best if you enhanced your contributions, limitations, underscore the scientific value-added of your paper, and the applicability of your findings/results and future study in this session.

6. PLOS authors have the option to publish the peer review history of their article (what does this mean?). If published, this will include your full peer review and any attached files.

Reviewer #1: No

Reviewer #2: **Yes: **KASHIF ABBASS

---

## [Author Response · Author response to Decision Letter 0]

8 Sep 2022

Dear Editors and Reviewers:

Thank you for your letter and for the reviewers’ comments concerning our manuscript entitled “Analysis of urban residents' online car-hailing journey characteristics based on the MGWR model” (ID: PONE-D-22-20651). Those comments are all valuable and very helpful for revising and improving our paper, as well as the important guiding significance to our researches. We have studied comments carefully and have made correction which we hope meet with approval. Revised portion are marked in red in the paper. The main corrections in the paper and the responds to the reviewer’s comments are as fllowing.

We sincerely hope the new version meet with the publication standard of plos one. Please let us know if anything else is needed and we look forward to hearing from you soon.Thank you and best regards.

Yours sincerely,

Yan Cao

Corresponding Author:

Yongzhong Tian

Associate Professor

School of Geographical Sciences, Southwest University

Chongqing 400715, China

Tel/Fax: +86-17729619166

Email: tyzlf@swu.edu.cn

Responds to the reviewer #1’s comments

Point 1: The paper is poorly expressed in language. Pay attention to the mixed use of past and present tense, especially, in the same sentence.

Response: We are very sorry that our incorrect writing has caused you problems in reading. Our manuscripts have been edited by English language editors before submission, but due to our negligence there may still be some errors in the text. We have carefully checked English throughout the manuscript and made a comprehensive revision in the new manuscript. 

Point 2: There are too many abbreviations in the paper, which seriously affects the reading as well as the understanding of the contents.

Response: We are so sorry that the abbreviations in the manuscript have caused confusion in your reading. Unnecessary abbreviations in the manuscript were reduced and we retained the abbreviations of the study periods, such as L227-L228.

Point 3: The introduction needs to be reorganized to highlight the value and meaning of the research. Too much discussion of variables selection is not appropriate for the introduction.

Response: We reorganized the introduction and divided it into Introduction and Literature review. In the introduction, we emphasized the value and significance of this study, and summarized previous studies on variable selection and influence model in the literature review.

Point 4: The discussion of the results is mainly on the sign of the variables, and the analysis of the causes of the effects is not in-depth. The number of variables of interest are too much, resulting in unfocused findings.

Response: Thank you for your valuable comments. The results section were reorganized. We analysed the spatial and temporal characteristics of residential trips, then compared the main variables in the results of each model and explored the causes, and in the discussion section the key variables that passed all the model tests were analysed.

Point 5: The conclusion is brief and the policy recommendations lack relevance and application.

Response: We have expanded the concluding section. The novelty and value of the findings of this study were emphasised, and each of the findings in our research process was analysed for its applicability.

Point 6: The title is not specific enough and does not indicate the core contents of the study. The introduction section points out "the purpose of this study is to examine the spatiotemporal relationship between online cab-hailing and the urban built environment", but the title is a study of "characteristics", which is confusing. It is recommended to revise it to cover and reflect your core research contents.

Response: It is really true as Reviewer suggested that the title does not cover and reflect our core research contents. We thought carefully about the content of our title and revised it, changing the original title to " Impact of built environment on residential online car-hailing trips: based on MGWR model ".

Point 7: Unify the expressions of the core terminology in the whole paper.

(1) Such as: car-hailing or cab-hailing? Please explain to me the difference between the two concepts and why you chose this one and not the other? If the two concepts refer to the same content, please unify the expressions in the text directly.

(2) DiDi has an English version of its website, please check the English spelling of DiDi. Please unify the expressions in the whole text.

Response: We are very sorry for neglecting to unify the expressions of the core terminology in the whole paper.

(1) Car-hailing, cab-hailing or ride-hailing have the same concept. The overarching idea of them is that a customer hires a driver to take them exactly where they need to go, something accomplished by hailing a taxi from the street, calling up a car service on the phone,or virtually hailing a car and driver from an app. We unified our expressions and car-hailing was used in the manuscript.

(2) We checked and confirmed their correct English writing of DiDi on DiDi's official website. DiDi was used in the whole text.

Point 8: L67: What are the others referring to?

Response: Thank you very much for your valuable comments. The 5D's principles of built environment and taxi trips were developed by Cervero and Ewing in 1997 and 2010 respectively. We have corrected our writing errors and removed the phrase “as well as others”.

Point 9: L141-L147:

(1) There is actually another peak(13:00-14:00) in Figure 2, why is this not discussed?.

(2) MP and EP? In the case of abbreviations for the first time, please use the whole words.

(3) The presentation of the content is illogical, and the English expression level is very poor.Please have your paper copy edited by a native speaker. Pay attention to the mixed use of past and present tense, especially, in the same sentence. Be consistent.

Response: We have re-written the part:

(1) In the corrected paragraph, we have described the peaks for this time period (13:00-14:00) and the reasons for choosing the morning and evening peaks and late night as the study periods in the manuscript.

(2) Abbreviations have been correcteded in the text.

(3) We have reorganized this section and carefully corrected tense errors.

Point 10: Section 2.2 contains too much contents and the logic is not clear enough. Please rearrange the logic of Section 2 and divide it into a proper chapters.

Response: Considering the Reviewer’s suggestion,we have rearranged the Section 2. We divided the section 2 into " Online car-hailing trips data " and " Built environment variables ". We put it in the section "Study Area" about the content of the research grid division.

Point 11: L229: (1)Where are the Tables A1 and A2? (2)Poor English proficiency and confusing use of tenses in this paragraph.

Response: 

(1) Tables A1 and A2 are actually S1 and S2 Tables, they are placed at the end of the manuscript.

(2)We are so sorry for our poor English writing. The tense errors have been carefully corrected.

Point 12: L237~L248: (1)Lack of standardization in table designing. (2)Please correspond to Table 1 for the order of variables in Table 2. (3)What is the difference between R-square and R2 in Table 2? (4)L241: EP? OR EPP? (5)It is not appropriate to discuss the conclusions of Table 4 here. (6)Please explain the meaning of AICc in the note part of Table 2. (7)Please explain on what software or algorithm these results are based?

Response: 

(1) All tables in the text have been redesigned to meet the requirements of the journal.

(2) Thanks for your comments. We have reordered the variables in Table 2.

(3) We are very sorry about the use of the wrong abbreviation. R-square is the residual sum of squares in previous versions. In the revised manuscript, we update it to RSS to avoid misunderstandings.

(4) In this place it mainly refers to the EPP model, which has been rewritten in the text.

(5) We have reformatted this section and removed inappropriate parts of it.

(6) Thank you for your sincere and careful comments. We have explained the meaning of AICc in the note part of Table 2.

(7) The calculations of the MGWR model in this paper are based on the Spatial Analysis Research Center (SPARC) at Arizona State University, USA. The MGWR2.2 software is developed by SPARC (https://sgsup.asu.edu/SPARC), the maps are produced using ArcGIS 10.6 software. This section has been added to the text in lines 319 to 321.

Point 13: Some of the details:

L124: I don't think it's appropriate for the contents to appear in this sentence: "rich and diverse internal land use variables (variables?)".

Line126: I don't think the contens of the brackets, (such as Jinniu and Chenghua), make sense.

L140, L147: A period is missing at the end of the sentence.

L159: [Please check that I have understood this section correctly.]?

L194: distributions

L197: Table1, the explained variables are usually placed before the explanatory variables.

The Table 4 and Table 5 are misplaced. When discussing tables, please place the table below the text.

Conclusion: Please note the use of punctuation in the whole test.

Response: It is really true as Reviewer suggested that there are some of the details of the errors that occurred during the writing process.

(1) L124: Thank you for your kind comments. We have changed the “variables”to “types”.

(2) L126: We have rewritten the sentence and listed all the study areas covered in our study area.

(3) L140, L194, L197: We have carefully corrected punctuation errors and grammatical errors in our full text.

(4) L159: We are very sorry for our negligence. The irrelevant sentence in the new manuscript have been deleted.

(5) L197: The order of explanatory variables and explained variables were reorganized in Table 1. Considering your valuable suggestion, we have repositioned Tables 4 and 5 and placed them below the corresponding text.

We tried our best to improve the manuscript and made some changes in the manuscript. These changes will not influence the main content and framework of the paper. And here we did not list the changes but marked in red in revised paper.We appreciate for your warm work earnestly, and hope that the correction will meet with approval.

Once again, thank you very much for your comments and suggestions.

Responds to the reviewer #2’s comments

Point 1: The manuscript needs language, grammar, and syntactic editing. The English language usage should be checked by a fluent English speaker.

Response: We are very sorry that our incorrect writing has caused you problems in reading. Our manuscripts have been edited by English language editors before submission, but due to our negligence there may still be some errors in the text. We have carefully checked English throughout the manuscript and made a comprehensive revision in the new manuscript. 

Point 2: The more suitable title should be selected for the article

Response: It is really true as Reviewer suggested that the title does not cover and reflect our core research contents. We thought carefully about the content of our title and revised it, changing the original title to " Impact of built environment on residential online car-hailing trips: based on MGWR model ".

Point 3: Also, find some new related references which should add to the 1st and 2nd parts.

Response: Thank you for your valuable comments. Some new related references were added in the Introduction and Literature review.

Point 4: Must add much more explanations and interpretations for the results, which are not enough. It is suggested to compare the results of the present research with some similar studies which is done before (more justification is needed).

Response: We have made correction according to your valuable comments. The results section were reorganized. We analysed the spatial and temporal characteristics of residential trips, then compared the main variables in the results of each model and explored the causes, and in the discussion section the key variables that passed all the model tests were analysed.

Point 5: Please make sure your conclusions section underscores the scientific value-added of your paper and the applicability of your findings/results, as indicated previously. Please revise your conclusion part into more detail. It would be best if you enhanced your contributions, limitations, underscore the scientific value-added of your paper, and the applicability of your findings/results and future study in this session.

Response: We have re-written this part according to your thoughtful and sincere comments. The novelty and value of the findings of this study were emphasised, and each of the findings in our research process was analysed for its applicability. To illustrate the limitations of this study and future work, we added the " Limitations and future work " section.

We tried our best to improve the manuscript and made some changes in the manuscript. These changes will not influence the main content and framework of the paper. And here we did not list the changes but marked in red in revised paper.We appreciate for your warm work earnestly, and hope that the correction will meet with approval.

Once again, thank you very much for your comments and suggestions.

---

## [Decision Letter · Decision Letter 1]

17 Oct 2022

PONE-D-22-20651R1Impact of built environment on residential online car-hailing trips: based on MGWR modelPLOS ONE

Dear Dr. Tian,

Thank you for submitting your manuscript to PLOS ONE. After careful consideration, we feel that it has merit but does not fully meet PLOS ONE’s publication criteria as it currently stands. Therefore, we invite you to submit a revised version of the manuscript that addresses the points raised during the review process. As highlighted by the reviewer, there are only minor issues that need to be revised. You can find the revision requests in the Reviewer's report.

We look forward to receiving your revised manuscript.

Kind regards,

Eda Ustaoglu, PhD

Academic Editor

PLOS ONE

Journal Requirements:

Reviewers' comments:

Reviewer's Responses to Questions

**Comments to the Author**

1. If the authors have adequately addressed your comments raised in a previous round of review and you feel that this manuscript is now acceptable for publication, you may indicate that here to bypass the “Comments to the Author” section, enter your conflict of interest statement in the “Confidential to Editor” section, and submit your "Accept" recommendation.

Reviewer #1: All comments have been addressed

2. Is the manuscript technically sound, and do the data support the conclusions?

Reviewer #1: Yes

3. Has the statistical analysis been performed appropriately and rigorously? 

Reviewer #1: N/A

4. Have the authors made all data underlying the findings in their manuscript fully available?

Reviewer #1: Yes

5. Is the manuscript presented in an intelligible fashion and written in standard English?

Reviewer #1: Yes

6. Review Comments to the Author

Reviewer #1: The paper has been greatly improved, both in terms of language and logic of expression. The authors' efforts regarding the revision of the second edition are worthy of recognition. In order to improve the quality of the paper, I would like to make the following comments:

1. L39-L47: The discussion of the advantages of DiDi data should be placed in the literature review section. I suggest putting it after L133, the discussion on data limitations. Start a separate paragraph.

2. In the introduction section, three difficulties are presented (L48-L63) and this research has been improved in three ways (L64-L76). Please reorganize the presentation of this part so that the research difficulties and your contribution are presented separately and correspond to each other.

3. L64-L65: This is the content of your research, not the purpose. You can directly indicate in the opening section that the contribution of this paper is reflected in the following three areas.

4. L73-L76: A discussion of the model results would be inappropriate here.

5. L76-L80: This is the purpose of your research, please state it on a separate line. Add your content chapter arrangement in this part.

6. L134-L139: The expression is fragmented and not well connected to the literature. Please integrate this section with the "contribution" in the introduction section and highlight your innovation in this section.

7. L239："MGWR uses different bandwidths for each variable, unlike the classical GWR model, which requires different levels of spatial smoothing for each variable;" I think what you are trying to convey is that MGWR can “requires different levels of spatial smoothing for each variable”, but the subject of the second half of this sentence is GWR. Please verify if the meaning of this sentence matches what you are trying to express.

8. It is best not to have a figure in the discussion section about the results. Please put Figure 5 and related contents in another appropriate place. It is suggested that Chapter 3 be discussed in several subsections, and the discussion on Figure 5 could be a subsection of Chapter 3.

Other comments:

9. ! Please be responsible for your own paper, read it carefully and correct all details of the mistakes.

(1) L46, L318: A period is missing at the end of the sentence.

(2) The content of the form should be standardized: Note that the number of decimal points in the tables is the same for the same category of data, so that it doesn't look cluttered.

(3) Make sure that the first letter of the same content is the same in different tables.

(4) Please unify the citation of all references, such as L215, etc.

(5) L226-L228: The letters of the formula and the explanation should correspond in case. j is a lowercase letter.

(6) Please check the text for all grammatical errors. E.g. L267: S1 and S2 Tables show …..; L271: shows?

10. L123: “Second,…..”

11. L149: Two “study area” in one sentence is wordy. The former emphasizes the object of study and the latter emphasizes the area.

12. It is not necessary to emphasize "in this paper” or “in this study". E.g. L163 and L166.

13. L168: too many “and”

14. Use "(1) Variable 1." or other tag before each variable explanation so that it corresponds to your variables selections in the first paragraph.

15. L235-L237: It should be placed before table1.

16. In order to make it easier for readers to distinguish, I suggest writing “Morning peak 1 model (pick-ups), Morning peak 2 model (drop-offs) ’’in Table 2, so as in Table 3-5.

7. PLOS authors have the option to publish the peer review history of their article (what does this mean?). If published, this will include your full peer review and any attached files.

Reviewer #1: No

---

## [Author Response · Author response to Decision Letter 1]

26 Oct 2022

Dear Editors and Reviewers:

Thank you for your letter and for the reviewers’ comments concerning our manuscript entitled “Impact of built environment on residential online car-hailing trips: based on MGWR model” (ID: PONE-D-22-20651R1). Those comments are all valuable and very helpful for revising and improving our paper, as well as the important guiding significance to our researches. We have studied comments carefully and have made correction which we hope meet with approval. Revised portion are marked in red and blue in the paper. The main corrections in the paper and the responds to the reviewer’s comments are as fllowing.

We sincerely hope the new version meet with the publication standard of plos one. Please let us know if anything else is needed and we look forward to hearing from you soon.Thank you and best regards.

Yours sincerely,

Yan Cao

Corresponding Author:

Yongzhong Tian

Associate Professor

School of Geographical Sciences, Southwest University

Chongqing 400715, China

Tel/Fax: +86-17729619166

Email: tyzlf@swu.edu.cn

Responds to the reviewer #1’s comments

Point 1: L39-L47: The discussion of the advantages of DiDi data should be placed in the literature review section. I suggest putting it after L133, the discussion on data limitations. Start a separate paragraph.

Response: Thank you for your valuable comments. We reorganized the advantages of DiDi data and put it in the Literature review.

Point 2: In the introduction section, three difficulties are presented (L48-L63) and this research has been improved in three ways (L64-L76). Please reorganize the presentation of this part so that the research difficulties and your contribution are presented separately and correspond to each other.

Response: This section has been reorganized. We have separated the difficulties and improvements so that they correspond to each other.

Point 3: L64-L65: This is the content of your research, not the purpose. You can directly indicate in the opening section that the contribution of this paper is reflected in the following three areas.

Response: Thank you for your sincere opinion. We modified this part of the statement and pointed out our contribution at the beginning.

Point 4: L73-L76: A discussion of the model results would be inappropriate here.

Response: It is really true as Reviewer suggested that this discussion are not appropriate here. After careful consideration, we delete this part of the content.

Point 5: L76-L80: This is the purpose of your research, please state it on a separate line. Add your content chapter arrangement in this part.

Response: We changed the position of the research purpose, and the chapter arrangement was placed after this paragraph.

Point 6: L134-L139: The expression is fragmented and not well connected to the literature. Please integrate this section with the "contribution" in the introduction section and highlight your innovation in this section.

Response: Thank you for your valuable comments. We reorganized this section and highlighted the contribution and innovation of our research.

Point 7: L239: "MGWR uses different bandwidths for each variable, unlike the classical GWR model, which requires different levels of spatial smoothing for each variable;" I think what you are trying to convey is that MGWR can “requires different levels of spatial smoothing for each variable”, but the subject of the second half of this sentence is GWR. Please verify if the meaning of this sentence matches what you are trying to express.

Response: Thank you very much for your valuable comments. We are sorry for the mistake due to our negligence. The correct meaning of this sentence is as you point out. We have corrected the error in this sentence.

Point 8: It is best not to have a figure in the discussion section about the results. Please put Figure 5 and related contents in another appropriate place. It is suggested that Chapter 3 be discussed in several subsections, and the discussion on Figure 5 could be a subsection of Chapter 3.

Response: We put Figure 5 in Chapter 4 and divided Chapter 4 into different sections depending on the content.

Point 9: ! Please be responsible for your own paper, read it carefully and correct all details of the mistakes.

(1) L46, L318: A period is missing at the end of the sentence.

(2) The content of the form should be standardized: Note that the number of decimal points in the tables is the same for the same category of data, so that it doesn't look cluttered.

(3) Make sure that the first letter of the same content is the same in different tables.

(4) Please unify the citation of all references, such as L215, etc.

(5) L226-L228: The letters of the formula and the explanation should correspond in case. j is a lowercase letter.

(6) Please check the text for all grammatical errors. E.g. L267: S1 and S2 Tables show …..; L271: shows?

Response: We are very sorry for neglecting these details in the whole paper. We read the manuscript carefully and correct it for detail mistakes.

Point 10: L123: “Second,…..”

Response: Thank you for your kind comments. We have changed the sentence.

Point 11: L149: Two “study area” in one sentence is wordy. The former emphasizes the object of study and the latter emphasizes the area.

Response: To reduce wordy, we rewrote the sentence according to the context.

Point 12: It is not necessary to emphasize "in this paper” or “in this study". E.g. L163 and L166.

Response: We carefully checked the full manuscript and reduced the number of words "in this paper" and "in this study".

Point 13: L168: too many “and”

Response: Thank you for your kind comments. We rewrote the sentence to avoid too many “and”.

Point 14: Use "(1) Variable 1." or other tag before each variable explanation so that it corresponds to your variables selections in the first paragraph.

Response: Thank you for your valuable comments. We added some variable labels like " (1) POIs " before each category of variable explanation.

Point 15: L235-L237: It should be placed before table1.

Response: We changed the position of this section and placed before Table 1.

Point 16: In order to make it easier for readers to distinguish, I suggest writing “Morning peak 1 model (pick-ups), Morning peak 2 model (drop-offs) ’’in Table 2, so as in Table 3-5.

Response: Based on your valuable comments, we have changed the corresponding contents of Tables 2 to 5.

We tried our best to improve the manuscript and made some changes in the manuscript. These changes will not influence the main content and framework of the paper. And here we did not list the changes but marked in red and blue in revised paper.We appreciate for your warm work earnestly, and hope that the correction will meet with approval.Once again, thank you very much for your comments and suggestions.

---

## [Decision Letter · Decision Letter 2]

31 Oct 2022

PONE-D-22-20651R2Impact of built environment on residential online car-hailing trips: based on MGWR modelPLOS ONE

Dear Dr Tian,

Thank you for submitting your manuscript to PLOS ONE. After careful consideration, we feel that it has merit but does not fully meet PLOS ONE’s publication criteria as it currently stands. Therefore, we invite you to submit a revised version of the manuscript that addresses the points raised during the review process.

There are only minor issues need to be revised as recommended by the reviewer. Please check the reviewer's report for these issues. Your manuscript will be accepted after revisions are completed.

We look forward to receiving your revised manuscript.

Kind regards,

Eda Ustaoglu, PhD

Academic Editor

PLOS ONE

Journal Requirements:

Reviewers' comments:

Reviewer's Responses to Questions

**Comments to the Author**

1. If the authors have adequately addressed your comments raised in a previous round of review and you feel that this manuscript is now acceptable for publication, you may indicate that here to bypass the “Comments to the Author” section, enter your conflict of interest statement in the “Confidential to Editor” section, and submit your "Accept" recommendation.

Reviewer #1: All comments have been addressed

2. Is the manuscript technically sound, and do the data support the conclusions?

Reviewer #1: Yes

3. Has the statistical analysis been performed appropriately and rigorously? 

Reviewer #1: Yes

4. Have the authors made all data underlying the findings in their manuscript fully available?

Reviewer #1: Yes

5. Is the manuscript presented in an intelligible fashion and written in standard English?

Reviewer #1: Yes

6. Review Comments to the Author

Reviewer #1: This article has met the criteria for acceptance. After modifying the following details, it is recommended to accept. (Authors are advised to check the article carefully for detailed errors)

L62 and L63 are suggested in the same paragraph

L115: two periods at the end of the sentence.

L115 and L116 should be in the same paragraph

L248: S1 and S2 show

L334: Table 5, there's an extra vertical line to the right of the table

Others…

Also, check the use of Spaces in the whole paper， there should be only one space between words.

7. PLOS authors have the option to publish the peer review history of their article (what does this mean?). If published, this will include your full peer review and any attached files.

Reviewer #1: No

---

## [Author Response · Author response to Decision Letter 2]

1 Nov 2022

Dear Editors and Reviewers:

Thank you for your letter and for the reviewers’ comments concerning our manuscript entitled “Impact of built environment on residential online car-hailing trips: based on MGWR model” (ID: PONE-D-22-20651R2). Those comments are all valuable and very helpful for revising and improving our paper, as well as the important guiding significance to our researches. We have studied comments carefully and have made correction which we hope meet with approval. Revised portion are marked in red in the paper. The main corrections in the paper and the responds to the reviewer’s comments are as fllowing.

We sincerely hope the new version meet with the publication standard of plos one. Please let us know if anything else is needed and we look forward to hearing from you soon.Thank you and best regards.

Yours sincerely,

Yan Cao

Corresponding Author:

Yongzhong Tian

Associate Professor

School of Geographical Sciences, Southwest University

Chongqing 400715, China

Tel/Fax: +86-17729619166

Email: tyzlf@swu.edu.cn

Responds to the reviewer #1’s comments

Point 1: L62 and L63 are suggested in the same paragraph

L115: two periods at the end of the sentence.

L115 and L116 should be in the same paragraph

L248: S1 and S2 show

L334: Table 5, there's an extra vertical line to the right of the table

Others…

Response: Thank you for your valuable comments. We are sorry for the mistake due to our negligence. First, as you suggested, we combined L62 and L63, L115 and L116 into the same paragraph, respectively. Second, We corrected errors in L115, L248 and L334. In addition, we have read the full manuscript again carefully and corrected the detail errors, especially the use of Spaces between words.

We tried our best to improve the manuscript and made some changes in the manuscript. These changes will not influence the main content and framework of the paper. And here we did not list the changes but marked in red in revised paper. We appreciate for your warm work earnestly, and hope that the correction will meet with approval. Once again, thank you very much for your comments and suggestions.

---

## [Editor Report · Decision Letter 3]

3 Nov 2022

Impact of built environment on residential online car-hailing trips: based on MGWR model

PONE-D-22-20651R3

Dear Dr. Tian,

We’re pleased to inform you that your manuscript has been judged scientifically suitable for publication and will be formally accepted for publication once it meets all outstanding technical requirements.

Kind regards,

Eda Ustaoglu, PhD

Academic Editor

PLOS ONE
---

## [Editor Report · Acceptance letter]

8 Nov 2022

PONE-D-22-20651R3 

Impact of built environment on residential online car-hailing trips: based on MGWR model 

Dear Dr. Tian:

I'm pleased to inform you that your manuscript has been deemed suitable for publication in PLOS ONE. Congratulations! Your manuscript is now with our production department. 

Kind regards, 

on behalf of

Dr. Eda Ustaoglu 

Academic Editor

PLOS ONE